# An Improved qFibrosis Algorithm for Precise Screening and Enrollment into Non-Alcoholic Steatohepatitis (NASH) Clinical Trials

**DOI:** 10.3390/diagnostics10090643

**Published:** 2020-08-28

**Authors:** Wei-Qiang Leow, Pierre Bedossa, Feng Liu, Lai Wei, Kiat-Hon Lim, Wei-Keat Wan, Yayun Ren, Jason Pik-Eu Chang, Chee-Kiat Tan, Aileen Wee, George Boon-Bee Goh

**Affiliations:** 1Department of Anatomical Pathology, Singapore General Hospital, Singapore 169856, Singapore; lim.kiat.hon@singhealth.com.sg (K.-H.L.); wan.wei.keat@singhealth.com.sg (W.-K.W.); 2Duke-NUS Medical School, National University of Singapore, Singapore 169857, Singapore; jason.chang@singhealth.com.sg (J.P.-E.C.); tan.chee.kiat@singhealth.com.sg (C.-K.T.); goh.boon.bee@singhealth.com.sg (G.B.-B.G.); 3Department of Pathology, Beaujon Hospital Paris Diderot University, 92110 Clichy, France; pierre.bedossa@liverpat.com; 4Peking University People’s Hospital, Peking University Hepatology Institute, Beijing Key Laboratory of Hepatitis C and Immunotherapy for Liver Diseases, Beijing International Cooperation Base for Science and Technology on NAFLD Diagnosis, Beijing 100044, China; liufeng@pkuph.edu.cn; 5Hepatopancreatobiliary Center, Beijing Tsinghua Changgung Hospital, Tsinghua University, Beijing 100084, China; weilai@mail.tsinghua.edu.cn; 6Institute for Precision Medicine, Tsinghua University, Beijing 100084, China; 7HistoIndex Pte. Ltd., Singapore 139955, Singapore; ren.yayun@choututech.com; 8Department of Gastroenterology and Hepatology, Singapore General Hospital, Singapore 169608, Singapore; 9Department of Pathology, National University Hospital, Singapore 119074, Singapore; aileen_wee@nuhs.edu.sg; 10Yong Loo Lin School of Medicine, National University of Singapore, Singapore 117597, Singapore

**Keywords:** NAFLD, NASH fibrosis, qFibrosis

## Abstract

Background: Many clinical trials with potential drug treatment options for non-alcoholic fatty liver disease (NAFLD) are focused on patients with non-alcoholic steatohepatitis (NASH) stages 2 and 3 fibrosis. As the histological features differentiating stage 1 (F1) from stage 2 (F2) NASH fibrosis are subtle, some patients may be wrongly staged by the in-house pathologist and miss the opportunity for enrollment into clinical trials. We hypothesized that our refined artificial intelligence (AI)-based algorithm (qFibrosis) can identify these subtle differences and serve as an assistive tool for in-house pathologists. Methods: Liver tissue from 160 adult patients with biopsy-proven NASH from Singapore General Hospital (SGH) and Peking University People’s Hospital (PKUH) were used. A consensus read by two expert hepatopathologists was organized. The refined qFibrosis algorithm incorporated the creation of a periportal region that allowed for the increased detection of periportal fibrosis. Consequently, an additional 28 periportal parameters were added, and 28 pre-existing perisinusoidal parameters had altered definitions. Results: Twenty-eight parameters (20 periportal and 8 perisinusoidal) were significantly different between the F1 and F2 cases that prompted a change of stage after a careful consensus read. The discriminatory ability of these parameters was further demonstrated in a comparison between the true F1 and true F2 cases as 26 out of the 28 parameters showed significant differences. These 26 parameters constitute a novel sub-algorithm that could accurately stratify F1 and F2 cases. Conclusion: The refined qFibrosis algorithm incorporated 26 novel parameters that showed a good discriminatory ability for NASH fibrosis stage 1 and 2 cases, representing an invaluable assistive tool for in-house pathologists when screening patients for NASH clinical trials.

## 1. Introduction

Non-alcoholic fatty liver disease (NAFLD) is among the most common chronic liver diseases, affecting an estimated 25% of the general adult population worldwide [1,2,3,4]. NAFLD pathogenesis is complex and multifactorial and is closely linked to metabolic syndromes. It represents a spectrum of clinical phenotypes, ranging from simple steatosis to nonalcoholic steatohepatitis (NASH), a more aggressive form characterized by steatosis, active hepatocyte injury in the form of ballooning degeneration, lobular inflammation, and a propensity for fibrosis [4,5]. As there are no other non-invasive modalities that are comparable, histological evaluation of liver tissue remains critical in defining the various phenotypes accurately, particularly for NASH [6,7,8]. Critically, the stage of fibrosis is an important feature to be assessed for NAFLD, as it impacts on overall and liver-related mortality [9,10]. Currently, there is a myriad of clinical trials in progress that seek effective therapeutic agents [11]. However, the immense difficulty of the task at hand is illustrated by the many failed remedies, such as simtuzumab, selonsertib, emricasan, and insulin sensitizer MSDC-0602K, all of which were unable to meet the histological endpoint of fibrosis improvement [12].

Many of the clinical trials have focused on patients with NASH stages F2 and F3 fibrosis. For some trial sites, patients are screened for eligibility based on historical biopsy data. Potentially, some samples that are initially deemed suitable by in-house pathologists, due to them being perceived as showing F2 or F3 fibrosis, fail enrollment after subsequent review by a central pathologist reader, who re-assigns them as F1 or F4 fibrosis. Conversely, some potential subjects with F2 fibrosis are misclassified by in-house pathologists to have lower-stage fibrosis and hence are not considered for trials. These discordant impressions reflect the real-world setting and represent substantial numbers of subjects being enrolled erroneously. It can be challenging to differentiate subtle changes in fibrosis with the routine use of traditional histochemical techniques that highlight collagen, even in the hands of expert hepatopathologists. This may represent missed opportunities to enroll suitable patients into clinical trials.

Second-harmonic generation (SHG) microscopy (qFibrosis) has been described as a precise and robust method for assessing fibrosis in NASH since it provides more granularity toward staging fibrosis on a continuous score rather than the traditional semi-quantitative method [13,14]. We hypothesize that qFibrosis can identify the subtle differences in F1 and F2 fibrosis patients and increase the accuracy of patient enrollment into NASH clinical trials by exploring signature qFibrosis parameters that arise because of discordant readings between in-house and central read pathologists.

## 2. Materials and Methods

### 2.1. Study Population and Samples

Liver tissue from 160 adult patients (≥18 years old) with biopsy-proven NAFLD/NASH were included in the current study. Cases were recruited from Singapore General Hospital (SGH), Singapore, and Peking University People’s Hospital (PKUH), China. Patients with liver disease of other etiologies, such as alcoholic or drug-induced liver disease, autoimmune liver disease, viral hepatitis, and cholestatic or genetic liver diseases were excluded. This study was approved by the Ethics Committee of both the participating hospitals (CIRB reference 01 July 2015/2527).

Thick sections 4–5 μm in size from the 160 samples were used for SHG imaging and subsequently stained with hematoxylin and eosin (H&E) and a connective tissue stain, such as Masson trichrome or Victoria blue for the confirmatory pathologist assessment of fibrosis. The minimum length of liver biopsy specimens was 10 mm. Specimens deemed inadequate in terms of size or unsatisfactory for technical/other reasons by the expert hepatopathologists were excluded from the study.

### 2.2. Image Acquisition

Unstained liver biopsy tissues of the 160 samples were imaged using the commercially available Genesis system (HistoIndex Pte Ltd., Singapore 139955, Singapore), where SHG microscopy was used to visualize collagen and two-photon excited fluorescence (TPEF) microscopy was utilized for visualization of the pertinent cell structures. The laser settings were as previously described [15]. In essence, the system involves a laser passed through a pulse compressor (for group velocity dispersion) and an acoustic-optic modulator (for power attenuation), following which, is routed by a dichroic mirror through an objective lens to the tissue sample. This creates a TPEF emission and an SHG signal, which are collected and processed for detection. The Genesis system is a closed system, where only the objective lens is adjustable (10×, 20×, and 40×), while all other components, such as the condenser, dichroic mirror, collection filters, and collection frame rate are not adjustable.

The samples were laser-excited at 780 nm, SHG signals were recorded at 390 nm, and TPEF signals were recorded at 550 nm. Images were acquired at 20× magnification with 512 × 512 pixel resolutions, and each image tile had a dimension of 200 × 200 μm. Multiple adjacent image tiles were captured to encompass the whole tissue areas in each slide.

### 2.3. Assessment of Fibrosis Stage

Of the 160 samples, 52 samples came from PKUH and 108 samples came from SGH. Liver fibrosis was scored as F0 to F4 using the Brunt staging system, with stage 4 fibrosis (F4) indicating cirrhosis [16]. The distribution of the samples according to their fibrosis stages are shown in Table 1.

### 2.4. Differences between One-Time Readings and Real-Time Readings

There are important differences between the two institutions in the way the fibrosis stages of liver biopsy samples are read. The PKUH samples were read by a single expert hepatopathologist (AW) over 1 week for a separate study.

In contrast, the SGH samples were based on historical histopathology reports by in-house non-specialty pathologists, which is the “real-world” setting in most NASH trial recruitment centers. As the cases were also recruited over 10 years from 2005 to 2015, there was also significant temporal variation.

### 2.5. Consensus Read

To ascertain the true fibrosis stage of the liver biopsy samples, a consensus read was organized, with all the sample cases assessed by two expert hepatopathologists (Aileen Wee, Wei-Qiang Leow), one of whom (Aileen Wee) was the same hepatopathologist who read the PKUH cases. The consensus read was organized twelve months after the initial read of the PKUH samples.

During the consensus read, liver fibrosis was scored in accordance with the NASH Clinical Research Network (NASH-CRN) criteria, which evaluates fibrosis similarly to the Brunt staging system, except that stage 1 is subdivided into stage 1a (delicate zone 3 perisinusoidal fibrosis), stage 1b (dense zone 3 perisinusoidal fibrosis), and stage 1c (periportal fibrosis only; seen in pediatric NASH) [17].

### 2.6. Creation of Periportal Parameters for the Refined qFibrosis Algorithm

Following the consensus read data, 40.6% (65/160) of the liver biopsy samples were re-classified. This was in part due to the recognition of portal fibrosis by the expert hepatopathologists. According to the NAFLD Activity Score (NAS), F2 fibrosis is defined by the presence of both pericellular/perisinusoidal fibrosis and periportal fibrosis.

In the earlier versions of the qFibrosis algorithm, although we found an increasing trend in the SHG B-index for Brunt stages 0, 1, and 2, there was considerable overlap and thus statistical significance could not be achieved [13]. We postulated that the qFibrosis algorithm under-recognized the presence of portal/periportal fibrosis. Hence, a periportal region, defined as a 100 μm radial margin around the portal tract, was incorporated into the qFibrosis algorithm (Figure 1). As a result of the new periportal region, an additional 28 parameters were added into the refined qFibrosis algorithm (Table 2). With the new periportal regions, the previous perisinusoidal (PS) regions were altered, which resulted in changes to 28 perisinusoidal parameters (Table 3).

### 2.7. Validation of the Improved qFibrosis Algorithm

We used this improved qFibrosis algorithm on the cases designated as F1 and F2 after the consensus read in order to elucidate the most statistically significant parameters with discriminatory capabilities.

### 2.8. Statistical Analysis

The Wilcoxon rank-sum test was performed to evaluate the difference between the parameters of F1 and F2 cases. The statistical significance level was set at *p* < 0.05. Statistical analyses were done with MATLAB R2019b (MathWorks, Inc., Natick, MA, USA).

## 3. Results

The consensus read by the two hepatopathologists resulted in changes to the original fibrosis stages in 30.7% (16/52) of PKUH liver samples and 45.3% (49/108) of SGH liver samples (Table 4).

In the analysis of the performance of the 28 new periportal parameters and 28 altered perisinusoidal parameters, we designated the cases that changed between F1 to F2/3 as one group (*n* = 17) and compared it against the group of samples that did not change their fibrosis stage (*n* = 30). There were interesting differences between the two institutions (Table 5).

In the PKUH column, none of the 56 parameters showed any differences when we compared the two groups. In striking contrast, the SGH column showed statistical differences in 28 out of the 56 parameters when we compared the two groups. Of these 28 parameters, 20 parameters were from the new periportal region and 8 parameters were from the altered perisinusoidal regions.

We postulate that the differences between the two institutions were due to the one-time reading method of PKUH as opposed to the real-time reading method of SGH. As the SGH samples were read by non-specialty pathologists, fibrosis in the periportal and perisinusoidal regions may be under-appreciated, whereas the one-time read by the expert histopathologist for the PKUH samples allowed for the upfront histopathological identification of finer forms of fibrosis. This underscores the need for a central histopathology read or an assistive tool with similar abilities prior to recruitment. The significant temporal variation of 10 years in the SGH cohort likely contributed to the reporting variability.

We further validated the discriminatory ability of these 28 parameters by comparing them between the consolidated F1 and F2 cases that did not change stages after the consensus read of both institutions and were deemed as true F1 and F2 cases. Twenty-six out of the 28 parameters showed significant differences, further substantiating the excellent discriminatory ability of these parameters (Table 6).

## 4. Discussion

Although the NAFLD activity score (NAS) from the NASH Clinical Research Network (NASH-CRN) is the most commonly used index in NASH clinical trials, there are only a few studies that have evaluated the operating properties of the NAS [18,19]. These studies demonstrated weak-to-moderate inter-observer agreement for the diagnosis of NASH between community general pathologists and expert hepatopathologists, highlighting a real-world issue in the applicability of such scoring systems. Despite the use of various histochemical techniques, including collagen fiber stains, such as Masson trichrome, or elastic fiber stains, such as Victoria blue, there is still considerable inter- and intra-pathologist variation [20]. Serendipitously, it was also this real-world issue that has allowed us to discover our discriminatory parameters.

R.K. Pai, in his critical review of NAFLD scoring systems, highlights the inability of the staging system to fully capture the degree of perisinusoidal fibrosis that can be present at all stages [21]. Although perisinusoidal fibrosis defines F1 disease in adults, it is not evaluated at the higher stages of fibrosis. A system that can detect and quantify this pattern of fibrosis will have value in predicting liver-related outcomes.

Many studies have shown that fibrosis is the most important feature when predicting adverse liver-related outcomes and mortality in NAFLD [7,22,23,24]. Most trials have designated endpoints that include a change in the total NAS score, a ≥ 2-point reduction in the steatosis-activity-fibrosis (SAF) score, a ≥ 2-point reduction in the NAS with no worsening of fibrosis, and a ≥ 1-stage decrease in fibrosis [25]. The endpoint in fibrosis has been hard to achieve, partly due to the lack of a linear relationship between the fibrosis stage and collagen deposition, and partly due to the limited range of fibrosis stages in the scoring systems. Hence, utilizing a system that can perform quantitative measurements of collagen and measure perisinusoidal fibrosis at all stages will improve the evaluation of this essential outcome.

In recent years, many studies have utilized analytic imaging tools to quantify the amount of collagen. Through machine learning and deep neural learning networks, such automated measurements have shown considerable correlations to outcomes [26,27,28,29,30,31,32]. As a result, clinical trials are exploring the utility of such technologies in the definition of their trial endpoints. As with all new technology, a lead-time is needed for maturation, and it is this continuous refinement of the algorithms anchored by biological relevance that will allow the technology to flourish. As F2 fibrosis starts at the periportal regions, we have refined our algorithmic parameters and incorporated additional novel parameters that allow us to identify patients with F2 fibrosis with more accuracy. Future algorithms should also assimilate similar conceptual changes. 

Quantitative fibrosis changes are small during the early stages of fibrosis progression. Such minute changes may be hard to discern, even with histochemical techniques. SHG and TPEF microscopy techniques allow for the visualization and quantification of early collagen. In our earlier study, the SHG B-index was able to differentiate advanced fibrosis (Brunt stage 3 and 4) from no or mild fibrosis (Brunt stage 0 and 1) [13]. Although there was considerable overlap between Brunt stages 0, 1, and 2, we noted an increasing trend in the SHG B-index. 

Our hypothesis that an algorithm with a better discriminatory ability for F1 and F2 cases lies in a focused look at the periportal region was proven correct by the discovery of 28 parameters, where 20 of them pertain specifically to the collagen changes in the periportal region. Furthermore, we were able to validate 26 of these 28 parameters when we used them to differentiate F1 from F2 cases in both institutions. These parameters provided the basis for the refinement of qFibrosis and were targeted at increasing the discriminatory ability of the algorithm in the early fibrosis stages, accurately stratifying F1 and F2 cases. As NASH clinical trial results are highly dependent on accurate F2 patient inclusion and F1 patient exclusion, we hope that this tool will provide better patient selection, increase enrollment, and lead to a breakthrough in the search for a NASH therapeutic option. 

There are limitations to this system. The first is that liver core biopsy tissue is still required. As the risk of an invasive procedure is still not negated, non-invasive blood and imaging-based biomarkers will always have the upper hand [33,34]. The widespread use of transient elastography is a testament to the clinical preference of non-invasive tools. The amount of tissue in an adequate core biopsy represents only 1/50,000 of the entire organ, and as with all biopsies, there is the issue of sampling bias [35,36]. Although statistical significance was found for almost all the parameters, we are mindful that the cohort that led to this discovery was small. Thus, we seek to validate our findings with a larger external cohort, which we are currently in the midst of coordinating. The external cohort validation will also overcome the potential intra-observer bias in the evaluation of the PKUH cohort. 

An aim for future research is to be able to predict the progression or regression of disease. The Beijing Classification developed by N.D. Theise for patients with chronic viral hepatitis is prophetic and future refinements of the technology should work toward this aim of prediction [37,38]. In addition, as we recognize that response to an improvement in steatohepatitis, such as decreased ballooning degeneration, steatosis, and portal inflammation, may also indicate therapeutic efficacy, both fibrosis and disease activity should be measured endpoints, as both are markers of therapeutic efficacy [39]. A combined system, such as qFIBS, addresses that need with a single technology [14].

## 5. Conclusions

The refined qFibrosis algorithm incorporated 28 additional parameters (20 periportal and 8 perisinusoidal), of which 26 novel parameters showed a good discriminatory ability for NASH fibrosis stage 1 and 2 cases, representing an invaluable assistive tool for in-house pathologists when screening patients for NASH clinical trials.

## Figures and Tables

**Figure 1 diagnostics-10-00643-f001:**
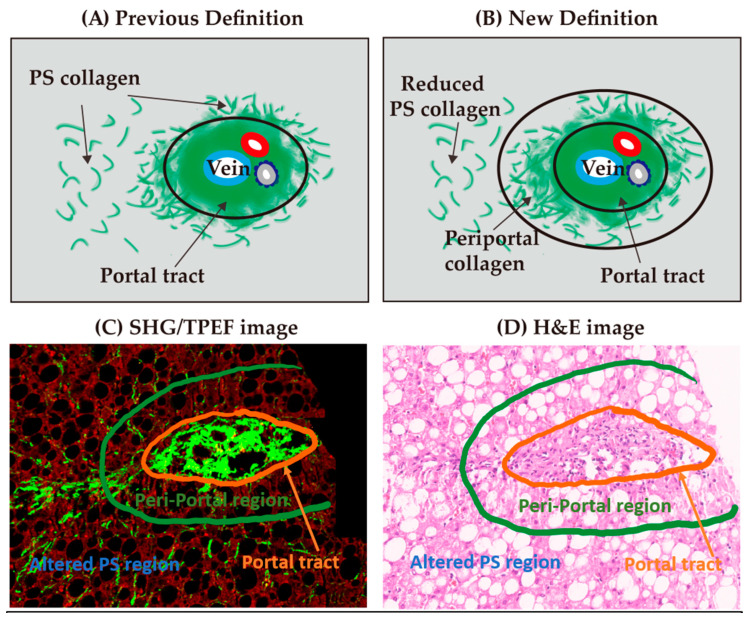
(**A**,**B**) Creation of a new periportal region with altered perisinusoidal (PS) collagen parameters. (**C**) Second-harmonic generation (SHG)/two-photon excited fluorescence (TPEF) image of the periportal region (in green) and portal tract (in orange). (**D**) Hematoxylin and eosin (H&E)-stained image of the periportal region.

**Table 1 diagnostics-10-00643-t001:** Distribution of samples according to non-alcoholic steatohepatitis (NASH) fibrosis stages before the consensus read.

Peking University People’s Hospital (PKUH) (*n* = 52)		Singapore General Hospital (SGH) (*n* = 108)		
Stage	Number (%)	Stage	Number (%)	
F0	17 (33%)	F0	17 (16%)	
F1	14 (27%)	F1	34 (31%)	
F2	6 (12%)	F2	11 (10%)	
F3	9 (17%)	F3	21 (19%)	
F4	6 (12%)	F4	25 (23%)	

**Table 2 diagnostics-10-00643-t002:** List of newly created periportal parameters.

Periportal Parameters	Descriptions
**%Periportal**	The percentage of fibers in the periportal region
**%PeriportalAgg**	The percentage of aggregated fibers in the periportal region
**%PeriportalDis**	The percentage of distributed fibers in the periportal region
**#StrPeriportal**	The total number of fibers in the periportal region
**#ShortStrPeriportal**	The number of short fibers in the periportal region
**#LongStrPeriportal**	The number of long fibers in the periportal region
**#ThinStrPeriportal**	The number of thin fibers in the periportal region
**#ThickStrPeriportal**	The number of thick fibers in the periportal region
**StrAreaPeriportal**	The total area of fibers in the periportal region
**StrLengthPeriportal**	The total length of fibers in the periportal region
**StrWidthPeriportal**	The total width of fibers in the periportal region
**#StrPeriportalAgg**	The total number of aggregated fibers in the periportal region
**#ShortStrPeriportalAgg**	The number of short aggregated fibers in the periportal region
**#LongStrPeriportalAgg**	The number of long aggregated fibers in the periportal region
**#ThinStrPeriportalAgg**	The number of thin aggregated fibers in the periportal region
**#ThickStrPeriportalAgg**	The number of thick aggregated fibers in the periportal region
**StrAreaPeriportalAgg**	The total area of aggregated fibers in the periportal region
**StrLengthPeriportalAgg**	The total length of aggregated fibers in the periportal region
**StrWidthPeriportalAgg**	The total width of aggregated fibers in the periportal region
**#StrPeriportalDis**	The total number of distributed fibers in the periportal region
**#ShortStrPeriportalDis**	The number of short distributed fibers in the periportal region
**#LongStrPeriportalDis**	The number of long distributed fibers in the periportal region
**#ThinStrPeriportalDis**	The number of thin distributed fibers in the periportal region
**#ThickStrPeriportalDis**	The number of thick distributed fibers in the periportal region
**StrAreaPeriportalDis**	The total area of distributed fibers in the periportal region
**StrLengthPeriportalDis**	The total length of distributed fibers in the periportal region
**StrWidthPeriportalDis**	The total width of distributed fibers in the periportal region
**#IntersectionPeriportal**	The number of intersections in the periportal region

**Table 3 diagnostics-10-00643-t003:** List of altered perisinusoidal parameters.

Perisinusoidal Parameters	Descriptions
**%RPS**	The percentage of fibers in the reduced perisinusoidal region
**%RPSAgg**	The percentage of aggregated fibers in the reduced perisinusoidal region
**%RPSDis**	The percentage of distributed fibers in the reduced perisinusoidal region
**#StrRPS**	The total number of fibers in the reduced perisinusoidal region
**#ShortStrRPS**	The number of short fibers in the reduced perisinusoidal region
**#LongStrRPS**	The number of long fibers in the reduced perisinusoidal region
**#ThinStrRPS**	The number of thin fibers in the reduced perisinusoidal region
**#ThickStrRPS**	The number of thick fibers in the reduced perisinusoidal region
**StrAreaRPS**	The total area of fibers in the reduced perisinusoidal region
**StrLengthRPS**	The total length of fibers in the reduced perisinusoidal region
**StrWidthRPS**	The total width of fibers in the reduced perisinusoidal region
**#StrRPSAgg**	The total number of aggregated fibers in the reduced perisinusoidal region
**#ShortStrRPSAgg**	The number of short aggregated fibers in the reduced perisinusoidal region
**#LongStrRPSAgg**	The number of long aggregated fibers in the reduced perisinusoidal region
**#ThinStrRPSAgg**	The number of thin aggregated fibers in the reduced perisinusoidal region
**#ThickStrRPSAgg**	The number of thick aggregated fibers in the reduced perisinusoidal region
**StrAreaRPSAgg**	The total area of aggregated fibers in the reduced perisinusoidal region
**StrLengthRPSAgg**	The total length of aggregated fibers in the reduced perisinusoidal region
**StrWidthRPSAgg**	The total width of aggregated fibers in the reduced perisinusoidal region
**#StrRPSDis**	The total number of distributed fibers in the reduced perisinusoidal region
**#ShortStrRPSDis**	The number of short distributed fibers in the reduced perisinusoidal region
**#LongStrRPSDis**	The number of long distributed fibers in the reduced perisinusoidal region
**#ThinStrRPSDis**	The number of thin distributed fibers in the reduced perisinusoidal region
**#ThickStrRPSDis**	The number of thick distributed fibers in the reduced perisinusoidal region
**StrAreaRPSDis**	The total area of distributed fibers in the reduced perisinusoidal region
**StrLengthRPSDis**	The total length of distributed fibers in the reduced perisinusoidal region
**StrWidthRPSDis**	The total width of distributed fibers in the reduced perisinusoidal region
**#IntersectionRPS**	The number of intersections in the reduced perisinusoidal region

**Table diagnostics-10-00643-t004a:** (**a**)

PKUH Samples			
Fibrosis Stages	Number of Cases	Changes after Consensus Scoring	
		Changes to Fibrosis Stages	Number (%)
F0	17	0	12 (71%)
		+1 ^#^	5 (29%)
F1	14	0	10 (71%)
		+1 ^#^*	4 (29%)
F2	6	0	6 (100%)
F3	9	−3 ^#^	1 (11%)
		−2 ^#^	2 (22%)
		−1 ^#^	2 (22%)
		0	3 (33%)
		1 ^#^	1 (11%)
F4	6	−1 ^#^	1 (17%)
		0	5 (83%)

**Table diagnostics-10-00643-t004b:** (**b**)

SGH Samples			
Fibrosis Stages	Number of Cases	Changes after Consensus Scoring	
		Changes to Fibrosis Stages	Number (%)
F0	17	0	11 (65%)
		+1 ^#^	6 (35%)
F1	34	−1 ^#^	10 (29%)
		0	17 (50%)
		+1 ^#^*	5 (15%)
		+2 ^#^*	2 (6%)
F2	11	−2 ^#^	2 (18%)
		−1 ^#^*	4 (36%)
		0	3 (27%)
		+1 ^#^	2 (18%)
F3	21	−3 ^#^	2 (10%)
		−2 ^#^	3 (14%)
		−1 ^#^	3 (14%)
		0	4 (19%)
		+1 ^#^	9 (43%)
F4	25	−1 ^#^	1 (4%)
		0	24 (96%)

**Table 5 diagnostics-10-00643-t005:** List of 28 significantly different parameters in the SGH cohort.

Statistically Significant Parameters	PKUH *p*-Value	SGH *p*-Value
**%PeriportalDis**	0.825	**0.022**
**#StrPeriportal**	0.414	**0.003**
**#ShortStrPeriportal**	0.710	**0.003**
**#LongStrPeriportal**	0.604	**0.009**
**#ThinStrPeriportal**	0.710	**0.024**
**#ThickStrPeriportal**	0.439	**0.004**
**StrLengthPeriportal**	1.000	**0.019**
**StrWidthPeriportal**	0.330	**0.004**
**#StrPeriportalAgg**	0.825	**0.005**
**#ShortStrPeriportalAgg**	0.940	**0.004**
**#LongStrPeriportalAgg**	0.484	**0.008**
**#ThickStrPeriportalAgg**	0.629	**0.005**
**StrWidthPeriportalAgg**	0.199	**0.006**
**#StrPeriportalDis**	0.825	**0.009**
**#ShortStrPeriportalDis**	0.940	**0.005**
**#ThinStrPeriportalDis**	0.940	**0.013**
**#ThickStrPeriportalDis**	0.710	**0.009**
**StrAreaPeriportalDis**	0.825	**0.022**
**StrLengthPeriportalDis**	0.604	**0.013**
**StrWidthPeriportalDis**	0.825	**0.019**
**#StrPS**	0.454	**0.049**
**#ThickStrPS**	0.454	**0.049**
**StrLengthPS**	0.635	**0.049**
**StrWidthPS**	0.539	**0.026**
**#StrPSAgg**	0.839	**0.042**
**#LongStrPSAgg**	0.733	**0.036**
**#ThickStrPSAgg**	0.733	**0.042**
**StrWidthPSAgg**	0.635	**0.036**

Note: The parameters with *p* < 0.05 are in bold.

**Table 6 diagnostics-10-00643-t006:** Validation of the discriminatory parameters.

Parameters	True F1 vs. True F2 Cases
**%PeriportalDis**	**0.011**
**#StrPeriportal**	**0.003**
**#ShortStrPeriportal**	**0.003**
**#LongStrPeriportal**	**0.005**
**#ThinStrPeriportal**	0.054
**#ThickStrPeriportal**	**0.005**
**StrLengthPeriportal**	**0.005**
**StrWidthPeriportal**	**0.011**
**#StrPeriportalAgg**	**0.005**
**#ShortStrPeriportalAgg**	**0.005**
**#LongStrPeriportalAgg**	**0.005**
**#ThickStrPeriportalAgg**	**0.008**
**StrWidthPeriportalAgg**	**0.011**
**#StrPeriportalDis**	**0.003**
**#ShortStrPeriportalDis**	**0.003**
**#ThinStrPeriportalDis**	0.054
**#ThickStrPeriportalDis**	**0.004**
**StrAreaPeriportalDis**	**0.011**
**StrLengthPeriportalDis**	**0.011**
**StrWidthPeriportalDis**	**0.008**
**#StrRPS**	**0.022**
**#ThickStrRPS**	**0.022**
**StrLengthRPS**	**0.022**
**StrWidthRPS**	**0.022**
**#StrRPSAgg**	**0.018**
**#LongStrRPSAgg**	**0.018**
**#ThickStrRPSAgg**	**0.022**
**StrWidthRPSAgg**	**0.022**

Note: The parameters with *p* < 0.05 are in bold.

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
