# Peer review of "An Improved qFibrosis Algorithm for Precise Screening and Enrollment into Non-Alcoholic Steatohepatitis (NASH) Clinical Trials"

_diagnostics, 2020, doi:10.3390/diagnostics10090643_

Round 1
Reviewer 1 Report
The manuscript presents a current real problem in medicine. Some NAFLD /NASH patients are labeled as having a higher or lower liver fibrosis stage by traditional biopsy and other methods are required for the proper classification of these individual. Treatment and inclusion of these patients in clinical trials depend on the correct staging of the disease.
The authors present a novel second harmonic generation microscopy (qFibrosis) to better classify the biopsies in patients with NAFLD/NASH.
The work is well presented and has its merits but there are some questions that I have that might need further clarification in the manuscript.
1) in line 83, the authors mention a minimum length of a liver biopsy of 10 mm. the question here is if the biopsies contained enough portal triads to make an appropriate diagnosis to start with and also, what the authors consider an appropriate biopsy for diagnosis.
2) In lines 123-124, the authors mention a consensus read, but this is carried out by one of the liver pathologists doing the first reading. Knowing that this pathologist has seen these biopsies before and that 40% of the biopsies were re-classified the question would be if there is the need for a third pathologist in the consensus read or other method to prevent intra-observer bias.
3) the newly created periportal parameters are only validated within the cohort of the biopsies that are reviewed. The question here is if the authors validated these newly created parameters with other biopsies or if these findings have been reproduced by other groups. Reproducibility becomes really important in this aspect.
4) lines 156-158 the authors admit that there are "interesting differences" between institutions when it comes to significant different parameters; how do the authors explain the differences that table 5 suggests that are more important in SGH?
5) in their discussion, the authors admit the limitations of the study and mentioned the invasive nature of the procedure and the technique that they are studying. No mention is made about elastography that is one of the most utilized technologies in the diagnosis and in some instances, follow up of patients with NAFLD/NASH. Is there any comment that they can make in this regard? Of course that this might not be capital for the aims of the manuscript.
Author Response
1) in line 83, the authors mention a minimum length of a liver biopsy of 10 mm. the question here is if the biopsies contained enough portal triads to make an appropriate diagnosis to start with and also, what the authors consider an appropriate biopsy for diagnosis.
Authors reply: The ideal liver biopsy should be 20mm long, 2mm wide and contain more than 11 portal tracts (Cholongitas E et al. Am J Clin Pathol. 2006;125(5):710-721.). However in practice, a more realistic criteria would be for the core biopsy to be 10mm long with approximately 6 to 10 portal tracts present for evaluation. As the number of portal tracts was not routinely reported, we decided to only use the length as minimum criteria. We are aware that in the setting of chronic hepatitis, the smaller the sample, the milder the disease (Colloredo G et al. J Hepatol. 2003 Aug; 39(2):239-44.). In the samples that met the minimum length criteria, our consensus review revealed that there was sufficient material for diagnosis. Samples that were assessed by the expert histopathologists to be inadequate were removed from the study (lines 84 to 85).
2) In lines 123-124, the authors mention a consensus read, but this is carried out by one of the liver pathologists doing the first reading. Knowing that this pathologist has seen these biopsies before and that 40% of the biopsies were re-classified the question would be if there is the need for a third pathologist in the consensus read or other method to prevent intra-observer bias.
Authors reply: We agree with the potential bias in our methodology. We felt that the bias is mitigated by the fact that the reads for the PKUH cohort was 12 months apart, and the time period would have diluted bias impression. We agree that this is a limitation of our study and have included this in our discussion (lines 234 to 236).
3) the newly created periportal parameters are only validated within the cohort of the biopsies that are reviewed. The question here is if the authors validated these newly created parameters with other biopsies or if these findings have been reproduced by other groups. Reproducibility becomes really important in this aspect.
Authors reply: We agree with the reviewer’s comment, and aim to validate our findings with a larger external cohort, which we are currently in the midst of coordinating (line 234).
4) lines 156-158 the authors admit that there are "interesting differences" between institutions when it comes to significant different parameters; how do the authors explain the differences that table 5 suggests that are more important in SGH?
Authors reply: We have elaborated further on our likely explanation for the differences (lines 164 to 171).
5) in their discussion, the authors admit the limitations of the study and mentioned the invasive nature of the procedure and the technique that they are studying. No mention is made about elastography that is one of the most utilized technologies in the diagnosis and in some instances, follow up of patients with NAFLD/NASH. Is there any comment that they can make in this regard? Of course that this might not be capital for the aims of the manuscript.
Authors reply: Thank you for the advice. We have made a mention of transient elastography in the manuscript (lines 228 to 230).
Reviewer 2 Report
The article of Leow et al., aimed to improve qFibrosis algorithm for the enrollment of Non-Alcoholic Steatohepatitis (NASH) patients from stage F1 to F2 in clinical trials. The article is interesting although there are some improvement that could be done before the publication:
Introduction: The authors must describe the histological feature of the various stage of NASH. Then qFibrosis is a relatively new method. Authors should describe it briefly in the introduction.
Methods and discussion: line 125. Authors should insert the extended name of the acronyms NAS and SAF.
Results and discussion: Authors must highlight clearly not just the differences found between the two centers (Table 5) but the parameters that were significative in both centers.
Author Response
Introduction: The authors must describe the histological feature of the various stage of NASH. Then qFibrosis is a relatively new method. Authors should describe it briefly in the introduction.
Authors reply: We have stated the spectrum of clinical phenotypes of NAFLD, and emphasised on the importance of presence of hepatocyte injury and inflammation in NASH (lines 47 to 48). We have also stated the stages of fibrosis and provided a reference to the staging system used (line 103). We have mentioned qFibrosis briefly in the introduction (lines 66 to 72), with further elaboration of the technique in methodology (lines 86 to 100).
Methods and discussion: line 125. Authors should insert the extended name of the acronyms NAS and SAF.
Authors reply: Thank you for highlighting. We have spelt out the acronyms (lines 127 and 197).
Results and discussion: Authors must highlight clearly not just the differences found between the two centers (Table 5) but the parameters that were significative in both centers.
Authors reply: We have elaborated further in an additional paragraph (lines164 to 171). The parameters that were significant in both centres are listed in Table 5, with the first 20 parameters associated with Peri-Portal features and the remaining 8 parameters associated with Peri-sinusoidal features. We postulate that these parameters reflect the under-recognition of fibrosis in these regions by the real-time read method of non-speciality pathologists.
Reviewer 3 Report
I have only minor suggestions for the authors:
- In the Introduction, please describe the key role of insulin resistance in NAFLD and the role of non invasive plasma biomarkers for predicting NAFLD and NASH; please consider these references (PMID: 31010049;PMID: 31169972) and comment these in the Introduction.
Author Response
- In the Introduction, please describe the key role of insulin resistance in NAFLD and the role of non invasive plasma biomarkers for predicting NAFLD and NASH; please consider these references (PMID: 31010049;PMID: 31169972) and comment these in the Introduction.
Authors reply: Thank you for the suggested references. We have linked NAFLD to metabolic syndrome (lines 46 to 47), and also included the references in our mention on non-invasive biomarkers (lines 228 to 230).